# Disconnected Body Representation: Neuroplasticity Following Spinal Cord Injury

**DOI:** 10.3390/jcm8122144

**Published:** 2019-12-04

**Authors:** Erik Leemhuis, Luigi De Gennaro, Mariella Pazzaglia

**Affiliations:** 1Department of Psychology, University of Rome “La Sapienza”, Via dei Marsi 78, 00185 Rome, Italy; luigi.degennaro@uniroma1.it; 2Istituto di Ricovero e Cura a Carattere Scientifico Fondazione Santa Lucia, Via Ardeatina 306, 00179 Rome, Italy

**Keywords:** phantom limb, spinal cord injury, neuroplasticity, somatotopy, somatosensory cortex, motor cortex, deafferentation, body representation

## Abstract

Neuroplastic changes in somatotopic organization within the motor and somatosensory systems have long been observed. The interruption of afferent and efferent brain–body pathways promotes extensive cortical reorganization. Changes are majorly related to the typical homuncular organization of sensorimotor areas and specific “somatotopic interferences”. Recent findings revealed a relevant peripheral contribution to the plasticity of body representation in addition to the role of sensorimotor cortices. Here, we review the ways in which structures and brain mechanisms react to missing or critically altered sensory and motor peripheral signals. We suggest that these plastic events are: (i) variably affected across multiple timescales, (ii) age-dependent, (iii) strongly related to altered perceptual sensations during and after remapping of the deafferented peripheral area, and (iv) may contribute to the appearance of secondary pathological conditions, such as allodynia, hyperalgesia, and neuropathic pain. Understanding the considerable complexity of plastic reorganization processes will be a fundamental step in the formulation of theoretical and clinical models useful for maximizing rehabilitation programs and resulting recovery.

## 1. Introduction

Body representation involves multiple neural sources of information. The primary somatosensory and motor cortices of the adult brain contain detailed maps of physical body topography that are thought to be adapted from interactions with the physical world [1].

This topographic organization allows us to predict major reorganization of the brain–body interaction, with changes in peripheral and central mechanisms following the loss of primary sensory and motor innervation.

Abnormalities in the cortical representation of the body, apparently arising from regions that have been completely denervated, is mostly documented in individuals with amputations. However relevant information on somatotopic remapping has also been recognized in peripheral disconnection (brachial plexus avulsion, BPA) [2] or spinal cord injury (SCI) [3]. Following input and output loss of limb BPA due to the traction/avulsion of cervical nerve roots from the spinal cord mimics the amputation, however, preserving the visual experience of the limb. Patients with BPA commonly report a remapping of sensation from the hand to the cheek [4], in the jaw and buccal region [5], or in the ear [6], presumably the same mechanisms that are responsible for phantom sensations in amputees. In SCI, the sensorimotor signaling between the brain and an entire portion of the body (and not only a specific limb) located below the level of lesion is interrupted permanently despite the physical body remaining unaltered. From the brain’s viewpoint, the somatotopic remapping of a complete SCI and amputation could be quite similar: in both cases, the rapid and complete loss of sensory input from a certain body part leads to a compensatory reorganization mechanism and phantom reduplication.

From a neuroscientific perspective, the mechanisms by which the brain transduces the flow of signals following peripheral disconnection are not yet understood. Moreover, despite the existence of several hypotheses, it is still unclear how the absence or critical impairment of sensory and motor signals in peripheral areas is compensated for by appropriate brain mechanisms and cortical territories. We took advantage of the clinical conditions of long-term sensorimotor loss to provide an ideal model for exploring brain reorganization, resulting in the physical preservation of the involved body region but with the complete loss of sensory and motor signals (in contrast to amputation). From this perspective, we consider alternative methods by which appropriate brain mechanisms and cortical territories reorganize the sensorimotor map after sensory and motor loss.

## 2. Disparities between Old and New in the Multiple-Scale Neural Plasticity of Brain-Body Reorganization Following Spinal Cord Injury (SCI)

Studies in both humans [7,8,9,10] and non-human primates [11,12,13] have detected substantial brain reorganization after SCI. A clear neural signature was identified in the primary somatosensory and motor cortices of primates after upper limb deafferentation with a medial spatial shift in facial representation (10–14 mm) toward the digit representation area [14]. Even if hand movement ability is mostly preserved, such deafferentation is sufficient to induce a similar expansion of the face representation, indicating that complete and incomplete lesions may promote similar displacement toward the adjacent face. These changes include the expansion of new facial connections into the deafferented hand region (in the primary somatosensory cortex [15] and the ventroposterior nucleus of the thalamus [16], spinal cord, and dorsal column nuclei [17]) as well as the transneuronal atrophy of deafferented structures [18].

Analogously, following the removal of sensory input from an upper limb nerve block in humans, participants reported swelling of the lips, providing evidence for a relationship between perceived body changes and areas neighboring the anesthetized region on the somatosensory map [19].

Furthermore, hemodynamic and electrophysiological changes in individuals with SCI indicate reorganization within and/or beyond the deafferented sensorimotor cortex. The increased activation of the somatosensory sites of the hand [20], finger [7], and tongue [21] shifting toward the deafferented area is associated with grey matter preservation and decreased fractional anisotropy [7]. Additionally, focal transcranial magnetic stimulation (TMS) has revealed the reorganization of corticospinal pathways, enlarged motor cortical representation maps, and shorter latencies in unaffected muscles represented immediately adjacent to the cortical deafferentation area. TMS also revealed reduced excitability and prolonged silent periods in muscles distal to the deafferented area [22], indicating that sensory motor cortical system reorganization predominantly occurs in neighboring zones after SCI. Moreover, specific “somatotopic interferences” of the face–hand homunculus are indicated by vivid “phantom” limb experiences in SCI that implicitly confirm the topographical remapping of the cortical territories following peripheral disconnection [23]. Despite the fact that selective somatotopic interference, cortical dynamics, and neuroplasticity based on competitive interaction from a normally innervated body are common and widely accepted (i.e., the phantom limb and anesthetization phenomena), there is no objective evidence supporting a neural topographic substitution in humans linked to peripheral loss, as has been observed in monkeys. Some of the discrepancies between human and animal neuroplasticity arise from interspecies biological differences and include variations in the way in which different species experience their bodies when interacting with the world. Therefore, the topographic reorganization of body parts is in response to activity and new skill acquisition [24]. Following peripheral loss, missing-hand territory can be flexibly assigned to body parts that share the same functional utility as the absent hand. After deafferentation, an animal is much more exposed to perform mouth actions than patients with SCI, leading to different neural topographic reorganization.

Additionally, in humans and animals, differences in the location and function of the corticospinal, rubrospinal, and reticulospinal tracts likely influence the activity of several motor clusters in the spinal cord, which innervate and control the limb muscles [25]. Finally, in individuals with injuries neurologically classified as “complete” [26], a rim of some intact peripheral tissue and fibers is spared [27], indicating the different potential of functional changes.

However, beyond these differences, many aspects remain unclear and controversial. Consistent with this classical view of expanding representation, it would be prudent to assume that the loss of functionality and reorganization should be reflected in the representation of body image; however, behavioral empirical evidence is controversial.

Recent studies on patients with SCI suggest that, despite the profound alteration of the dynamic organization of peripheral sensation and movement, permanent and extensive sensorimotor disconnection has minimal effects on the body templates (i.e., depictive body image measure), even after extensive periods of deafferentation [28,29]. Therefore, the body metric (i.e., the dimension and positions of key points on the body relative to each other) is not based on signals from any single receptor or afferent pathway but indicates the existence of multiple pathways for body representation. However, as posited by both theoretical [30,31] and quantitative [28,32,33,34] studies, the acquisition of wheelchair proficiency by SCI patients alters the metrical perceptions of the body, suggesting an interplay between the body template and online information from the periphery updated via successive visual and proprioceptive changes in body position. At the functional level, it is commonly assumed that the deprivation of peripheral neural signals can strengthen the internal somatosensory body processes in neighboring cortical zones after SCI. Indeed, following long-term sensory deafferentation of the lower limb, the advantage of normal proprioception in patients with paraplegia prevents the overwriting of proprioceptive information via visual capture, such as, for example, the proprioceptive drift classically found in the rubber hand illusion [35]. Under such simultaneous multisensory interplay, vision typically dominates proprioception and touch, but not in patients with paraplegia. Large-scale cortical reorganization in certain body areas may determine an enlarged and strengthened internal proprioceptive representation of a paraplegic’s hand that prevented the illusion from overwriting proprioceptive information regarding hand localization.

Additionally, multisensory stimulation can directly benefit the tactile and pain perception of deafferented fingers adjacent to the sentient fingers [36,37]. These findings implicitly suggest the functional advantages of cross-modal effects but dissociating the contributions of the cortical and subcortical mechanisms of neuroplasticity is challenging.

However, a more profound concern regarding the traditional remapping hypothesis is the extent to which results derived from brain plasticity studies in adult primates can be extended to human brain plasticity and whether the mechanisms involved extend beyond the intracortical level. 

Multiple research groups have challenged two main hypotheses: the primary role of the sensorimotor cortex in the reorganization process and the assumption that the residual ascending sensory pathways are fundamentally silent and play a limited role in recovery [38,39]. They suggest that future plasticity models should consider the observation that, after traumatic events, pre-injury sensorimotor representations can “survive” along with the newly formed connections. Recent advancements in material engineering and surgical techniques allowed for the implantation of microelectrodes, organized in dense arrays, directly on the brain surface in patients with a SCI. Using carefully modulated electrical micro-stimulation exactly on the primary sensorimotor regions, the researchers could elicit natural sensations that follow the expected somatotopic representation, indicating the remarkable stability of somatosensory cortical maps [39]. Furthermore, the ability to evoke strong and highly detailed phantom sensations by stimulating the stump supports the hypothesis of a preserved somatotopic organization. Functional cortical plasticity observed after deafferentation is likely to be a result of the reorganization of pathways originating from residual peripheral nerves and extending upward via the spinal cord, brainstem, and thalamus [38]. This notion suggests that the major changes in cortical organization that presumably reflect the adaptation of cortical maps to altered spinothalamic and spinocerebellar input occur due to the collateral sprouting of damaged or spared pathways, which form new neuronal tracts [40].

This view remains to be confirmed. Perhaps the only way to elucidate the intricate processes involved in post-deafferentation plasticity is to systematically incorporate other potentially associated variables into further investigations.

The effects of a SCI can present great variability, from slight reduced sensory and motor activity in an incomplete lesion to the abolished sensory and motor and interoceptive signals in a complete lesion, suggesting a degree of cortical reorganization of body which is highly variable [40]. Recent advances by resting state functional magnetic resonance imaging (MRI) indicate that the neural connectivity after acute and chronic SCI appears modified by indicating that reorganization may occur not only in the brain but also in the spinal cord [41,42]. Changes in neural connectivity along new pathways in the processing of sensorimotor information, in the spinal cord and brain cortex, occurring after SCI, support an integrated view and enhanced recognition of the bottom-up and top-down contributions [41,42]. While many anatomical and functional brain and spinal cord changes may be related to the extent of neurological injury and lesion severity, considering the cases of neurologically complete injury, a variety of additional factors are inherently involved in reorganization after deafferentation [43,44].

Time is a critical factor in experience-dependent plasticity, and longitudinal studies are required to understand the progressive development in process timing that, by their very definition, extend across different time scales. Combining a longitudinal approach with the use of a wider variety of techniques, such as magnetic resonance imaging, would facilitate the acquisition of a greater amount of information pertaining to cross-sectional studies, which—in the case of neurological lesions—exhibit high variability among their subjects. The displacement of cortical sensorimotor representations is minimal one year after SCI and is oriented toward deafferented representation depending on the severity of the SCI [22]. Traumatic events, such as spinal injuries, are linked to the development of secondary damage, including neurodegenerative processes in the medullar, subcortical, and cortical structures [45]. After inflammatory processes that are typical of the acute phase, a loss of myelin and accumulation of iron lead to progressive atrophy involving the spinal cord, corticospinal tract, cerebellum, thalamus, primary and secondary sensory cortices, and the cingulate cortex [45]. This neurodegeneration process stabilizes between 24 and 36 months-post-SCI, in conjunction with what, in many patients, corresponds with the cessation of clinical progress. Higher rates of degeneration are associated with more severe sensorimotor impairment and greater neuropathic pain intensity. Moreover, the extent of atrophy and demyelination at six months-post-SCI can predict a patient’s outcome at two years [45].

Age affects brain plasticity and functional reorganization, both of which are associated with characteristics that rely on processes that vary during a lifetime. Traumatic events, such as SCIs or limb loss, may have different clinical outcomes depending on the age of the patient. Age-related factors influence the recovery process at the molecular level and beyond. One example is a reduction in the concentration of antioxidant defenses that is typical in older subjects and often associated with an increase in reactive oxygen species, which, in turn, exacerbates inflammatory events [46]. As possible consequences, interactions between increased inflammatory responses, reduced neuronal plasticity, and reduced regenerative potential may negatively affect the recovery process.

Subjective experiences collected from individual patients are an important source of potentially useful information, particularly when coupled with objective recordings of neurophysiological activity. Moreover, the phenomenon of phantom sensations has driven research on bodily representations and cortical plasticity for several decades. Phantom sensations can be elicited by touching body parts that are “cortical neighbors” of the deafferented location. It is less well-known that phantom sensations can undergo topographical changes from the acute phase to the subsequent healing phases. SCI patients report phantom limb sensations similar to those that are experienced by amputees. The incidence of paraplegia phantom cases may be the same as amputation phantom cases but are reported less often because many phantoms may coincide with reported sensations and movement below the neurological level of injury of a limb that remains visibly intact, thus creating conflicting messages [47,48]. In contrast with amputees, the visual impression may then become the prevalent source of body information, yet lacking essential body information, sensorimotor but also interoceptive and proprioceptive post-lesional perception. Somatotopic patterns are not, however, uniformly reported after SCI. Several referred phantom sensations evoked by contact with body parts that were both neighboring and distant and not predicted to be adjacent in the cortex, suggest instead a non-somatotopic remapping that is projected to the cortex [23,47,49]. 

Subcortical, rather than cortical, plastic changes may provide the referred phantom sensations of non-adjacent cortical regions (i.e., chest regions following forearm contact) after SCI [23].

An assessment of the emergence and course of progression of phantom hand sensation after a brachial plexus injury in a patient enabled the investigation of an unclear aspect of the sensory somatotopic organization of the ear [50]. Typically, the phantom sensation of a missing or deafferented hand can be evoked by touching the corresponding cheek and lower part of the face, but, in the aforementioned patient, the phantom sensation was evoked via stimulation of the aural region innervated by a branch of the vagus nerve [6]. Interestingly, after a few weeks, the phantom sensation began to gradually migrate from the ear and generally localized on the lower part of the face, suggesting ongoing sensory reorganization [51].

These not necessarily somatotopic phenomenon in body peripheral disconnection suggest the presence of possible subcortical sensory interference and the importance of deafferentation as a model for body representation research.

Maladaptive plasticity is another process that warrants further investigation. It is widely considered to be the neural basis of the emergence and chronicity of phantom limb pain. The complete somatosensory deprivation and activity loss of afferent information from the missing limb or the area of the body below the SCI is widely considered to be a trigger of the maladaptive process. The amount of reorganization in the primary somatosensory cortex was found to correlate with the reported intensity of neuropathic pain following SCI [52]. However, specific morphometric changes occur in the cortico-spinal tracts of individuals with SCI with and without neuropathic pain, suggesting on-going cortical and subcortical reorganization related to pain modulation. Notably, recent research has suggested the possibility of a much more active role of damaged peripheral nerves in the persistent representation of the missing limb [53,54]. Higher levels of cortical activity in the somatosensory area associated with the missing limb correlate with both phantom pain and the preserved ability to feel and mentally move the phantom hand, suggesting a limited impact of maladaptive reorganization.

## 3. Conclusions

As this work indicates, the classical theory of cortical reorganization, although still dominant in the literature, does not adequately explain some of the most recent findings. Both subcortical structures and residual or ectopic peripheral nervous activity are gaining relevance in terms of shaping the representation of the body [38,39], creating a balanced picture of both top-down and bottom-up processes. Therefore, the unification of seemingly conflicting results would be a major achievement in the comprehension of the body–mind problem.

As in other fields, the need to integrate complex interactions into innovative research designs and clinical procedures is a natural consequence of the above-described aspects of the brain–body disconnection. Sensorimotor loss following SCI imposes a heavy burden on healthcare systems but represents an invaluable resource for basic and clinical research [55]. Recent technological advancements highlight the need for a multiple-scale joint research activity ranging from the molecular level to cognitive and human sciences [56]. The outcomes will help in the generation of new models aimed at reducing the growing discrepancies between engineering achievements and everyday life. A “side effect” of the great challenge in providing rehabilitation protocols for such patients includes questioning how to best use the brain–body (or body–brain) information. Clinical treatments aiming to increase and preserve the use of the on-line and off-line experiences of body sensation, sense of ownership, and sense of agency should be based on remapping inputs to the affected body parts into the preserved body parts. In particular, taking advantage of peripheral multisensory integration processes could enhance the perceptual experience and chronic pain management [36,37,57], thus improving life quality.

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
