# Peer review of "Disconnected Body Representation: Neuroplasticity Following Spinal Cord Injury"

_jcm, 2019, doi:10.3390/jcm8122144_

Round 1
Reviewer 1 Report
The authors provide a comprehensive review of current understanding of the physiological underpinnings of somatotopic reorganization following spinal cord injury. While this is an often studied topic following limb amputation (‘phantom limb’), less is known about this phenomenon following SCI and this topic is worthy of review.
Last sentence of first paragraph, should be “reorganize the sensorimotor map after sensory and motor loss”
Line 74-76 – expand a bit on how difference in human versus animal perceptions of the world (what are these differences, as best we understand them) can lead to different neural topographic reorganization following peripheral loss?
Line 83-84, define what is meant by ‘body template’ and ‘body metric’
Line 94 – describe the ‘rubber hand illusion’ and how it illustrates prevention of overwriting of proprioceptive information via visual capture in paraplegic patients
Line 106 – microstimulation of primary sensorimotor cortices?
Line 103-155 – describes both the stability and rigidity of primary sensorimotor topographies but at the same time discussion of the cortical plasticity from peripheral nerve reorganization? Need to clarify which theory is predominant or how they co-exist with eachother
Line 116-121 – Please describe the differences encountered with complete SCI. Also, seems to be a citation missing [xxxxx]
Line 127 – displacement of what?
Line 137 – How is atrophy and demyelination measured in these studies
Need to review phantom-type reorganization following SCI in more detail
Need to review in detail more the studies/evidence regarding neuroplasticity and the functional effects following SCI and draw a distinction to the peripheral nerve loss driven reorganization seen following limb loss
Line 178 – Should be ‘sensorimotor loss following SCI…’
Overall, would try and make the distinction between what is known about and the implications of neuroplasticity following limb amputation and SCI are. As well as describe in more detail the evidence behind what happens following SCI in more detail.
Author Response
REVIEWER 1
The authors provide a comprehensive review of current understanding of the physiological underpinnings of somatotopic reorganization following spinal cord injury. While this is an often studied topic following limb amputation (‘phantom limb’), less is known about this phenomenon following SCI and this topic is worthy of review.
We thank the reviewer for considering the topic interesting and for making useful suggestion for a more clear and complete paper. We have addressed all the amendments required by the reviewer in the revised version of the manuscript.
Last sentence of first paragraph, should be “reorganize the sensorimotor map after sensory and motor loss”
We thank the referee for pointing this out. The text on page 2 now reads as follows: reorganize the sensorimotor map after sensory and motor loss
Line 74-76 – expand a bit on how difference in human versus animal perceptions of the world (what are these differences, as best we understand them) can lead to different neural topographic reorganization following peripheral loss?
We completely agree with the reviewer. We have made this part more explicit:
Some of the discrepancies between human and animal neuroplasticity arise from interspecies biological differences and include variations in the way in which different species experience their bodies when interacting with the world. Therefore, the topographic reorganization of body parts is in response to activity and new skill acquisition [22]. Following peripheral loss, missing-hand territory can be flexibly assigned to body parts that share the same functional utility as the absent hand. After deafferentation, an animal is much more exposed to perform mouth actions than patients with SCI, leading to different neural topographic reorganization. Additionally, in humans and animals, differences in the location and function of the corticospinal, rubrospinal, and reticulospinal tracts likely influence the activity of several motor clusters in the spinal cord, which innervate and control the limb muscles [23]. Finally, in individuals with injuries neurologically classified as “complete” [24], a rim of some intact peripheral tissue and fibers is spared [25], indicating the different potential of functional changes.
Line 83-84, define what is meant by ‘body template’ and ‘body metric’:
We wanted to differentiate between a global representation of the body and the spatial information about single body parts needed to build it.
Text was rewritten to better explain the difference:
Recent studies on patients with SCI suggest that, despite profound alteration of the dynamic organization of peripheral sensation and movement, permanent and extensive sensorimotor disconnection has minimal effects on body templates (i.e. depictive body image measure), even after extensive periods of deafferentation [23,24]. Therefore, the body metric (i.e., the dimension and positions of key points on the body relative to each other) is not based on signals from any single receptor or afferent pathway, but indicates the existence of multiple pathways for body representation.
Line 94 – describe the ‘rubber hand illusion’ and how it illustrates prevention of overwriting of proprioceptive information via visual capture in paraplegic patients
This has been made clearer in the revised version of the manuscript. It now reads.
Indeed, following long-term sensory deafferentation of the lower limb, the advantage of normal proprioception in patients with paraplegia prevents the overwriting of proprioceptive information via visual capture, such as, for example, the proprioceptive drift classically found in the rubber hand illusion [33]. Under such simultaneous multisensory interplay, vision typically dominates proprioception and touch, but not in patients with paraplegia. Large-scale cortical reorganization in certain body areas may determine an enlarged and strengthened internal proprioceptive representation of a paraplegic’s hand that prevented the illusion from overwriting proprioceptive information regarding hand localization.
Line 106 – microstimulation of primary sensorimotor cortices?
This has been rephrased in:
Recent advancements in material engineering and surgical techniques allowed to implant microelectrodes, organized in dense arrays, directly on the surface of the brain in patients with a SCI. Using carefully modulated electrical micro-stimulation exactly on the primary somatosensory regions, the researchers could elicit natural sensations that follow the expected somatotopic representation indicating the remarkable stability of somatosensory cortical maps [31].
Line 103-155 – describes both the stability and rigidity of primary sensorimotor topographies but at the same time discussion of the cortical plasticity from peripheral nerve reorganization? Need to clarify which theory is predominant or how they co-exist with each other
In literature, the theory of “moving” cortical representation is widely diffused while the study of the role of residual peripheric neurologic activity is less explore and more recent. We share with the reviewer the need for more clarity, we added a paragraph to the conclusion section:
As this work indicates, the classical theory of cortical reorganization, although still dominant in the literature, does not adequately explain some of the most recent findings. Both subcortical structures and residual or ectopic peripheral nervous activity are gaining relevance in terms of shaping the representation of the body [38,39], creating a balanced picture of both top-down and bottom-up processes. Therefore, the unification of seemingly conflicting results would be a major achievement in the comprehension of the body–mind problem.
Line 116-121 – Please describe the differences encountered with complete SCI.
This has been made clearer in the revised version of the manuscript. It now reads:
The effects of a SCI can present great variability, from slight sensorimotor disturbances in an incomplete lesion to the loss of sensory signals and motor activity in a complete lesion, suggesting a degree of cortical reorganization of body highly variable [4].
Also, seems to be a citation missing [xxxxx]
We thank the reviewer for pointing out this. We now cite Gourab, K., & Schmit, B. D. (2010). Changes in movement-related β-band EEG signals in human spinal cord injury. Clinical Neurophysiology, 121(12), 2017-2023.
Line 127 – displacement of what?
The displacement of the cortical sensorimotor representation of the body. Text has been rephrased
Line 137 – How is atrophy and demyelination measured in these studies
Atrophy is measured via quantification of volumetric changes using computational neuroimaging approaches. Myelinization is measured with magnetization transfer saturation and iron content using effective transverse relaxation rate.
Need to review phantom-type reorganization following SCI in more detail
We thank the reviewer for this suggestion. We have added more information on the phantom in SCI :
SCI patients report phantom limb sensations similar to those that are experienced by amputees. The incidence of paraplegia phantom cases may be the same as amputation phantom cases but are reported less often because many phantoms may coincide with reported sensations and movement below the neurological level of injury of a limb that remains visibly intact, thus creating conflicting messages [44,45]. In contrast with amputees, the visual impression may then become the prevalent source of body information, yet lacking essential body information, sensorimotor but also interoceptive and proprioceptive post-lesional perception. Somatotopic patterns are not, however, uniformly reported after SCI. Several referred phantom sensations evoked by contact with body parts that were both neighboring and distant and nor predicted to be adjacent in the cortex suggest instead a non-somatotopic remapping that is projected to the cortex [23,44,46]. Subcortical, rather than cortical, plastic changes may provide the referred phantom sensations of non-adjacent cortical regions (i.e. chest regions following forearm contact) after SCI [23].
Need to review in detail more the studies/evidence regarding neuroplasticity and the functional effects following SCI and draw a distinction to the peripheral nerve loss driven reorganization seen following limb loss
We have added more information on differences :
Abnormalities in the cortical representation of the body, apparently arising from regions that have been completely denervated, is mostly documented in individuals with amputations. However relevant information on somatotopic remapping has also recognized in peripheral disconnection (brachial plexus avulsion, BPA) [2] or spinal cord injury (SCI) [3]. Following input and output loss of limb BPA due to the traction/avulsion of cervical nerve roots from the spinal cord, mimics the amputation, however preserving the visual experience of the limb. Patients with BPA commonly report a remapping of sensation from the hand to the cheek [4], in the jaw and buccal region [5], or in the ear [6], presumably the same mechanisms that are responsible for phantom sensations in amputees. In SCI, the sensorimotor signaling between the brain and an entire portion of the body (and not only a specific limb) located below the level of lesion is interrupted permanently despite the physical body remains unaltered. From the brain’s viewpoint, the somatotopic remapping of a complete SCI and amputation could be quite similar: in both cases, the rapid and complete loss of sensory input from a certain body part leads to compensatory reorganization mechanism and phantom reduplication.
Line 178 – Should be ‘sensorimotor loss following SCI…’
We thank the reviewer for pointing this out. Changed as suggested by the reviewer.
Overall, would try and make the distinction between what is known about and the implications of neuroplasticity following limb amputation and SCI are. As well as describe in more detail the evidence behind what happens following SCI in more detail.
We thank you and the referee for your thoughtful suggestions and insights, which have enriched the manuscript and produced a more balanced and better account of the research. With the made significant changes to points 2, 3, 4 and 12 we tried to give more information about SCI implications from the perspectives of neuroplasticity, comparison of human and animal models and subjective experience of the deafferented body. In the conclusions (answer to point 6) we have expanded the text to summarize the current dominant theoretical positions.
Reviewer 2 Report
In this review, the authors examine the question of how structures and brain mechanisms behave when sensory and motor peripheral signals are missing or critically altered, as in Spinal Cord Injury.
They suggest that these events are altered across multiple times, are age-dependent and related to a great degree to altered perceptual sensations during and after remapping of the deafferented peripheral area. Very importantly these plastic events may contribute to the development of secondary pathological conditions, including hyperalgesia and neuropathic pain.
These studies are aimed at understanding the complexities of the plastic reorganization processes that occur after SCI. Understanding how the reorganization occurs will be a fundamental step in the formulation of relevant clinical models aimed at improving rehabilitation programs and neuro recovery.
The paper is very clearly written, and the ideas are well articulated. On line 121 there is one missing reference. This article will be of great interest to your readership
Author Response
We thank the reviewer for the careful reading of our paper and for the positive evaluation. We have addressed all the amendments required by the reviewers in the revised version of the manuscript. On line 121, we now cited Gourab, K., & Schmit, B. D. (2010). Changes in movement-related β-band EEG signals in human spinal cord injury. Clinical Neurophysiology, 121(12), 2017-2023.